# Unique Features of the Tissue Structure in the Naked Mole Rat (*Heterocephalus glaber*): Hypertrophy of the Endoplasmic Reticulum and Spatial Mitochondrial Rearrangements in Hepatocytes

**DOI:** 10.3390/ijms23169067

**Published:** 2022-08-13

**Authors:** Valeriya Vays, Irina Vangeli, Chupalav Eldarov, Vasily Popkov, Susanne Holtze, Thomas Hildebrandt, Olga Averina, Dmitry Zorov, Lora Bakeeva

**Affiliations:** 1A.N. Belozersky Research Institute of Physico-Chemical Biology, Lomonosov Moscow State University, 119991 Moscow, Russia; 2Department of Reproduction Management, Leibniz-Institute for Zoo and Wildlife Research, Alfred-Kowalke-Str. 17, 10315 Berlin, Germany

**Keywords:** naked mole rat, hepatocytes, endoplasmic reticulum, mitochondria, electron microscopy

## Abstract

The reason for the exceptional longevity of the naked mole rat (*Heterocephalus glaber*) remains a mystery to researchers. We assumed that evolutionarily, *H. glaber* acquired the ability to quickly stabilize the functioning of mitochondria and endoplasmic reticulum (ER) to adjust metabolism to external challenges. To test this, a comparison of the hepatic mitochondria and ER of *H. glaber* and C57BL/6 mice was done. Electron microscopy showed that 2-months-old mice have more developed rough ER (RER) than smooth ER (SER), occupying ~17 and 2.5% of the hepatocytic area correspondingly, and these values do not change with age. On the other hand, in 1-week-old *H. glaber*, RER occupies only 13% constantly decreasing with age, while SER occupies 35% in a 1-week-old animal, constantly rising with age. The different localization of mitochondria in *H. glaber* and mouse hepatocytes was confirmed by confocal and electron microscopy: while in *H. glaber,* mitochondria were mainly clustered around the nucleus and on the periphery of the cell, in mouse hepatocytes they were evenly distributed throughout the cell. We suggest that the noted structural and spatial features of ER and mitochondria in *H. glaber* reflect adaptive rearrangements aimed at greater tolerance of the cellular system to challenges, primarily hypoxia and endogenous and exogenous toxins. Different mechanisms of adaptive changes including an activated hepatic detoxification system as a hormetic response, are discussed considering the specific metabolic features of the naked mole rat.

## 1. Introduction

The naked mole rat (*Heterocephalus glaber*) is a unique animal with specific features that stand beyond many recognized scientific paradigms. It has been shown that there are certain patterns in nature. In particular, in mammals, their lifespan usually correlates with other parameters of the life cycle, such as body weight or pregnancy period [1]. Indeed, the average lifespan of a mouse accounts for 1–2 years, while the naked mole rat, being comparable in body weight to a mouse (up to 35 g), has a maximum lifespan of up to 30 years [2]. In addition to such a uniquely long lifespan, the naked mole rat also features other exceptional characteristics, including high tolerance to infections, high regenerative capacity, and resistance to cancer and diabetes [3]. As a result, scientists’ focus on *H. glaber* features is understandable with the purpose to identify the mechanisms underlying its unusual high lifespan and resistance to external adverse environmental challenges as well as age-related diseases. 

Naked mole rats (*H. glaber*) areal covers underground labyrinths in arid and semi-arid regions of Kenya, Ethiopia, and Somalia [4,5], forming large colonies of up to 300 animals [5,6]. Holtze et al. have shown that the composition of air in naked mole rats’ labyrinths in wild nature is similar to that of normal air [7]. However, the authors note that, according to other studies [6,8,9], when all members of the colony rest in a common nest, CO_2_ levels can reach 7–10%, which is orders of magnitude higher than in normal air, which means that these animals have managed to adapt to hypoxia. In addition, naked mole rats have shown a minimal decline in reproductive and physiological parameters associated with aging [10], as well as an extremely low age-related increase in mortality [11,12], resistance to cancer [13,14] and hyperactivation of pathways involved in stress tolerance [6]. 

One of the possible mechanisms of a reduced pathological response to challenges, in particular to an oxidative challenge, is the discovered mechanism of mild uncoupling of oxidative phosphorylation in the mitochondria of a naked mole rat in all its tissues [15]. The essence of the protective mechanism of mild uncoupling is reduced to a decrease in the production of reactive oxygen species (ROS) due to activation of respiration caused by uncoupling and, consequently, a decrease in the degree of reduction of the components of the mitochondrial respiratory chain responsible for a single-electron transfer to the oxygen molecule, ultimately generating the superoxide anion radical. A decrease in the degree of mitochondrial coupling can be achieved not only by the direct generation of ion leak through the inner membrane of mitochondria but also due to the activation of ATPases of any nature, leading to a decrease in respiratory control [16,17,18,19]. Comparison of naked mole rats with short-lived mice has shown that aging is accompanied by inactivation of this mild depolarization mechanism almost in all tissues, which may lead to chronic and subsequently pathologic effects of toxic ROS levels for the organism. On the other hand, in murine liver cells, this mechanism becomes inactive immediately after birth, while in long-lived *H. glaber*, a moderate depolarization of the mitochondrial membrane in different organs, including the liver, continues for many years [15]. 

However, despite the potentialy lower generation of ROS in the naked mole rat, it has a much higher level of protein oxidation at early ages compared to mice [20,21]. This may indicate that the tissues of the naked mole rat are constantly in conditions that for other species would be considered a state of oxidative stress, even under conditions of low ROS generation but with inhibited elimination of oxidants and oxidative products. The tolerance to oxidative stress may become an evolutionarily achieved advantage, being a result of adaptation to oxidative stress ultimately yielding an extended lifespan. 

In addition, there is an age-independent high stability of proteins of *H. glaber* associated with increased proteasome activity in the liver [22,23]. Proteasomes are responsible for regulated protein degradation [24] and are generally considered an integral component in maintaining protein quality control, which, in its turn, can play a crucial role in maintaining health and longevity [25]. The distribution of proteasomes in the cell varies depending on the tissue and cell cycle stage, but in general, 60% to 90% of active proteasomes are present in the cytosol [26,27,28]. This subcellular fraction includes approximately 50% of proteasomes that interact with the cytosolic side of the endoplasmic reticulum. The remaining 10% to 40% of proteasomes are localized in the nucleus [26,27]. Notably, proteasome activity differs in the naked mole rat compared to other animals. Rodriguez et al. demonstrated that the level of proteasome activity in the liver of naked mole rats is more than two times higher than that in age-matched mice [28]. Obviously, any differences in the functioning of the most common liver cells, hepatocytes, described for the naked mole rat compared to other mammals, should be reflected at the ultrastructural level. However, biochemical studies of naked mole rat tissues are exclusively dominant, while no studies of the ultrastructure of *H. glaber* hepatocytes have been conducted yet. Taking into account significant differences in a number of biochemical parameters, the investigation of *H. glaber* hepatocytes is of great interest, as well as the comparison of their ultrastructure with that of murine hepatocytes. 

In this study, the ultrastructure of *H. glaber* hepatocytes was compared to that of C57BL/6 mice, and changes in the hepatocyte ultrastructure in naked mole rats of different ages (1 week, 3 years, 5 years, 7 years, and 11 years) were analyzed.

## 2. Results

The hepatic cell ultrastructure of C57BL/6 mice at the age of 2 months fully corresponded to classical concepts [23] (Figure 1a). Hepatocytes had an irregular hexagonal shape with indistinct angles in sections. In the central part of the cell, one or two rounded nuclei are typically localized. Generally, there was a sinusoidal pole facing the blood sinusoid and a biliary pole facing the bile duct. Notably, the internal space of hepatocytes was uniformly filled with cell organelles (Figure 1). Lysosomes and peroxisomes were found in the cytoplasm (Figure 1b). The endoplasmic reticulum (ER) was presented in its conventional forms: rough (granular) and smooth (agranular). Rough ER (RER) was present in the form of stacks (up to 20) of flattened cisterns that are parallel to each other and are distributed throughout the cytoplasm. Smooth ER (SER) was much less developed, with SER membranes localized close to the plasma membrane of hepatocytes. The ER membranes were in close contact with the membranes of the Golgi apparatus as well as with mitochondria, which were widely present in the cytoplasm (more than 1000 in one cell). 

The evaluation of the structure of the hepatocyte chondriome in C57BL/6 mice liver tissue was performed by image analysis of energized mitochondria in hepatocytes of non-fixed liver tissue. It was visualized using staining by a membrane potential-sensitive fluorescent probe, TMRE. The images demonstrate that the entire internal space of liver parenchymal cells, except for the nucleus area, is uniformly filled with energized round or oval-shaped mitochondria (Figure 2a,b). The resulting image fully corresponded to the electron microscopy images. 

The same approach, using thin sections of non-fixed hepatic tissue from 3-year-old *H. glaber,* demonstrated that, unlike murine hepatocytes, the population of energized mitochondria was not evenly distributed in the cytoplasm of liver parenchymal cells (Figure 2c,d). Along with the space occupied by the nucleus, it can be seen that in the central part of the cell, most of the cytoplasm is almost free of mitochondria, which are localized either around the nucleus or on the periphery of the cell (Figure 2d). There were no ultrastructural differences in the conformation of mitochondria in both types of rodents. Significant differences in the arrangement of energized mitochondria in *H. glaber* compared to mice might obviously reflect the characteristics of the hepatic cell ultrastructure of the naked mole rat and reflect different spatial energy demands across the cell compartments. 

Indeed, the electron microscopic examination showed that *H. glaber* has a completely different pattern of organelle distribution in hepatic cells compared to other animals. Figure 3a shows an overview photograph of the hepatocyte ultrastructure in the 3-year-old naked mole rat. A single nucleus was located in the center of a polygonal cell; the rest of the cytoplasm was densely occupied by cellular organelles. Numerous rounded or oval-shaped mitochondria were observed. However, in full compliance with the photometric analysis data, mitochondria were not evenly distributed throughout the cytoplasm but were located mostly around the nucleus and on the periphery of cells with rare exceptions. The rest of the internal space of hepatocytes was densely filled with a network of ER and SER rather than RER, represented by a system of tubules and cisterns (Figure 3a). At higher magnification, it can be seen that the RER is located mainly around the mitochondria, both in the nucleus area and on the cell periphery, with numerous osmiophilic granules of 12–15 nm in diameter, obviously ribosomes, located on its outer membrane (Figure 3b). The central part of the naked mole rat hepatocyte was densely filled with SER membranes. The observed pattern does not correspond to the conventional view on the structural organization of mammalian hepatocytes [29], with RER membranes contributing significantly to the entire ER.

Mammalian ER area changes occur only at the stage of early development. Thus, Kanamura et al. demonstrated [29] that in mice, the relative ER volume vs. cytoplasm volume usually increases in hepatocytes until 10–20 days after birth and then practically does not change until adulthood. On the other hand, such an increase in the ER area at the early developmental stages occurs due to increased SER area, while the RER area remains almost unchanged from birth and throughout life [30]. In order to trace changes in the ER area (RER and SER) with age, we conducted an electron microscopic and morphometric study with a thorough analysis of the dynamics of hepatocyte ER structure changes in *H. glaber* over time.

Figure 4a shows an overview image of the hepatic cell structure of the 1-week-old naked mole rat. The nucleus is located in the center of the cell. Numerous mitochondria, ER membranes, and other organelles are evenly distributed in the cytoplasm. Figure 4b represents a fragment of the hepatocyte of the naked mole rat at higher magnification. It can be seen that rat hepatocyte contains numerous stacks of the RER located both in the nucleus area and in the entire cell volume, in close contact with mitochondria. However, already at this age, a large area of cytoplasm free of mitochondria and RER is occupied by SER membranes. 

By the age of 7 years, the ratio of the RER and SER membranes in the naked mole rat hepatocytes continues to change with increasing relative SER volume (Figure 5). The overview image (Figure 5a) shows that almost the entire cell volume is occupied by the SER membrane, surrounded by numerous mitochondria entering the perinuclear area. At the same time, the relative area occupied by the RER membranes decreases compared to that of young animals. A fragment of the hepatocyte of the 7-year-old naked mole rat is shown at higher magnification in Figure 5b. It can be seen that RER no longer forms a stack of membranes: single cisterns are located between the mitochondria, and the entire internal space of the cells is filled with the SER membranes. 

By the age of 11 years, most of the internal space in the naked mole rat hepatocytes is occupied by the SER membranes (Figure 6a). Mitochondria are not evenly distributed in cells but are localized mainly in the perinuclear area and, by several groups, in the internal area of the cytoplasm. The RER in cells is represented by single cisterns located between mitochondria (Figure 6b). 

For morphometric analysis of the electron microscopy data, the proportion of the ER area relative to the total cell area was measured. The calculations were performed using 10 electron microphotographs of hepatocytes for each group of animals. We found that while in the hepatocyte of a 1-week-old naked mole rat, SER occupies 35.15% of the cell area (Figure 7, top), in a 7-year-old naked mole rat it becomes 43.95% and in 11-year-old animals it reaches 62.89%. Correspondingly, the relative volume of RER in hepatocytes in a 1-week-old naked mole rat is 12.84% (Figure 7, Bottom) with further decrease to 7.71% by the age of 5 years and remaining constant up to 11 years of age. To our knowledge, such observations of changes in hepatocytes’ endoplasmic reticulum have not been previously described. However, the morphometric analysis of the relative ER volume in murine hepatocyte yielded complete agreement with the classical concepts: at the age of 2 months, SER occupied 2.57% of the volume (Figure 7, top), while RER occupied 16.38% (Figure 7, Bottom), and these values almost did not change with age. 

## 3. Discussion

Our study demonstrates that hepatocytes of a naked mole rat have very specific ER and mitochondrial spatial arrangements as well as their ultrastructures if compared to murine hepatocytes. In the naked mole rat hepatocytes, the relative area occupied by SER area is more than 10-fold higher than in murine hepatocytes (Figure 7a). Another interesting result is the increasing area of the SER with age (Figure 7a). It was previously shown that in mice, SER area increases only within the period up to 10 days after birth, and after 20 days, this parameter does not significantly change [29]. On the contrary, our data from the naked mole rat shows that this area gradually increases throughout life. By 11 years of life, this value almost doubles compared to a 1-week-old animal (Figure 7a). 

The spatial distribution of the energy-competent mitochondrial population inside the hepatocyte in *H. glaber* differs from that in the mouse. Mitochondrial clustering in the perinuclear area and on the periphery of the cell in *H. glaber* may reflect different energy needs of the cellular compartments of the *H. glaber* hepatocyte. 

One of the possible mechanisms underlying *H. glaber* longevity is neoteny [3], i.e., “preservation of youthful characteristics in adulthood” [30]. The term “neoteny” was first applied to the naked mole rat in 1991 by R. Alexander [31]. The study run by V.P. Skulachev and co-authors provides information on 43 neotenic signs in the naked mole rats [3]. Possibly, the continuous increase in SER area for 11 years of life for this animal can be considered as another neotenic sign in *H. glaber*.

Thus, the remarkable distinctive feature identified regarding the naked mole rat hepatocytes is the extensively developed SER. It should be noted that there is no available data concerning the *H. glaber* hepatocyte ultrastructure, and our study was pioneering since most projects have been limited to the study of biochemical and molecular biological parameters rather than structural characteristics of the naked mole rat cells and tissues. 

It is known that the physiological role of the ER is to be involved in the neutralization of consumed and formed toxic substances, as well as bilirubin conjugation, steroid metabolism, biosynthesis of proteins secreted by the cell into tissue fluid, and direct participation in carbohydrate metabolism. 

Changes in the SER structure in a large number of studies of mammalian hepatocytes are shown under conditions of detoxification of toxic substances. SER is extensively enlarged during this process, filling the entire cytoplasm [32,33]. Jones and Fawcett have shown that phenobarbital administration in hamsters leads to significant growth of SER in hepatocytes [32]. Remmer and Merker demonstrated similar changes in the ultrastructure of rat and rabbit hepatocytes after injections of phenobarbital, nikethamide, and tolbutamide [33]. After removal of the toxic substances, the excessive ER network was destroyed through macroautophagy. The ultrastructural parameters of mammalian hepatocytes after exposure to poisons are similar to those observed in the intact naked mole rats. However, the naked mole rat has well developed SER in the hepatocytes already at the age of 1 week, and up to 11 years of life it constantly enlarges in the cytoplasm of hepatic cells. We can speculate that the observed hepatocyte ultrastructure features in the naked mole rat may reflect a chronically activated hepatic detoxification system as a hormetic response to intrinsic or extrinsic substances [34], which makes this animal distinct from other mammalian species. 

Another reason leading to such features of *H. glaber* hepatocyte ultrastructure could be a different system of degradation of used and initially misfolded protein compared to other animals. Many genes associated with protein homeostasis, including proteasome subunits, have recently been shown to be present at higher levels in *H. glaber* tissues compared to mice [35,36]. Despite the high level of oxidative damage even at a young age, the level of ubiquitinated proteins in these animals persists at a lower rate than in mice, both in young and elderly animals [20], which suggests reduced accumulation of damaged or misfolded proteins. These results may indicate the high efficiency of the ubiquitin-proteasome system in the naked mole rats. If the increased efficiency of this system is associated with the increased number of proteasomes in the naked mole rat, this may explain the increased size of SER in hepatocytes since it has been shown that approximately 50% of the proteasomes of the cell interact with the cytosolic side of the ER [26,27]. However, Rodriguez et al. demonstrated that, compared to mice, a similar amount of proteasome content is observed in the naked mole rats, but at the same time, their activity was higher [37]. This data shows that *H. glaber* has a highly efficient mechanism of protein degradation, constantly removing misfolded and damaged proteins throughout life. It is possible that the enlargement of the SER is associated with these processes, which are different from those of other animals.

SER enlargement may also result from specific metabolic features of this animal. In wild nature, naked mole rats have an underground lifestyle, forming large colonies of up to 300 animals [5]. Holtze et al. demonstrated that although the composition of air in naked mole rats’ burrows is similar to that of normal atmospheric air, naked mole rats regularly experience hypoxic episodes. This occurs while all colony members rest in a common nest, where they huddle together, or while digging narrow tunnels [7]. Park et.al., showed that the naked mole rat can withstand a complete absence of oxygen for 18 min and fully recover after such a challenge [9]. On the other hand, the tissue succinate-fumarate ratio, being the indicator of the tricarboxylic acid cycle, shows little change. The authors demonstrated that during anoxia, significantly increased levels of fructose and sucrose are observed in the blood, liver, and kidneys of the naked mole rat, while in mice this phenomenon has not been observed. Such high concentrations of fructose and sucrose can support energy metabolism under hypoxic conditions. Furthermore, Faulkes et al., have demonstrated that the glycogen content in the naked mole rat liver is noticeably (more than 10-fold) higher than in mice (4.1 ± 1.1 vs. 0.31 ± 0.08 mmoL/g; *p* < 0.001) [38]. 

Based on this data, we can conclude that in the natural habitat, the naked mole rat needs a constantly functioning, powerfully developed system for regulating the level of carbohydrates in the organism, capable of maintaining the energy supply of the animal ready to be exposed to hypoxia. The main sugar reserves capable of regulating blood sugar levels are stored in the liver, and it is SER that participates in the glycogen turnover The data obtained suggests that *H. glaber* hepatic cell ultrastructure features may be related to the specific metabolism characteristics of this animal. 

## 4. Materials and Methods

### 4.1. Animals

*Naked mole rats*. Five groups of naked mole rats (1-week-old, 3-, 5-, 7- and, 11-years-old) were used. Each group contained four animals. Naked mole rats were taken from colonies kept at the Leibniz-Institute for Zoo and the Wildlife Research (Berlin) and A.N. Belozersky research institute of Physico-Chemical Biology, Lomonosov Moscow State University. The naked mole rat colonies were kept in perplexing labyrinths. The temperature was maintained at 26–29 °C, and relative humidity 60–80%. Food was available ad libitum and included sweet potatoes, carrots, apples, fennel, groats with vitamins and minerals, and oat flakes. The experiments were approved by the Ethics Committee of the Landesamt für Gesundheit und Soziales, Berlin, Germany (#ZH 156; G 0221/12; T 0073/15).

*Mice*. Two groups of adult C57BI/6 male mice aged 2 months (n = 5) and 24 months (n = 3) were used. The animals were kept in individually ventilated cages (IVC) with 5–8 animals in each cage. Food and water were available ad libitum, with a 12/12 light cycle and an air temperature of 20–24 °C. Wood chips, Safe BK 8/15 (JRS, Germany), were used as bedding. 

### 4.2. Vital Liver Slices Imaging

Liver tissue (from segmentum hepatis anterius laterale dextrum VI) was excised and placed in the incubation medium (DMEM/F12 without sodium bicarbonate) to wash the blood. Tissue specimens were embedded in low-melting agarose (Thermo-Fischer Scientific, Waltham, MA, USA). Then, 70–100 μm thick sections were obtained with a Leica VT-1200s vibratome, washed with incubation medium (all procedures and incubation were done at 25 °C) and incubated for 30 min with 200 nM of tetramethyl rhodamine ethyl ester (TMRE, Thermo-Fisher Scientific, USA). Mitochondria in liver sections were imaged and analyzed using a LSM 710 inverted laser confocal microscope (Carl Zeiss, Jena, Germany) with a pinhole of 1 a.u. TMRE-loaded sections were evaluated with excitation light at 543 nm and emission >560 nm.

### 4.3. Electron Microscopy

Pieces of liver tissue (from segmentum hepatis anterius laterale dextrum VI) were excised and fixed with 3% glutaraldehyde solution (Sigma Aldrich, St. Louis, MO, USA) in 0.1 M phosphate buffer (pH 7.4) for 2 h at 4 °C. The pieces were further fixed with 1% osmium tetroxide for 1.5 h and dehydrated in an alcohol series with increasing ethanol concentrations of 50, 60, 70, 80, and 96% (70% ethanol contained 1.4% uranylacetate (Serva, Heidelberg, Germany)) to enhance contrast. After that, the samples were embedded in Epon812 epoxy resin. A series of sequential ultrathin sections were made with an ultra-microtome (Leica, Viena, Austria). Visualization was performed by a JEM1400 electron microscope (JEOL, Tokyo, Japan) equipped with a QUEMESA camera (Olympus, Center Valley, PA, USA), at an accelerating voltage of 100 kV and a beam current of 65 μA; images were processed with the software provided by the manufacturer (EMSIS GmbH, Muenster, Germany).

### 4.4. Morphometry and Statistical Analysis

For morphometric examination, ten electron microscopic images of hepatocytes at low magnification (×1500) for each group of animals were selected and analyzed. On each image, areas of granular and agranular endoplasmic reticulum were selected using graphics editing software (ImageJ) with a calculation of the area fraction for each type of ER in % of overall cell area. ANOVA and the t-test were used for statistical analysis, with a significance threshold of *p* = 0.05.

## Figures and Tables

**Figure 1 ijms-23-09067-f001:**
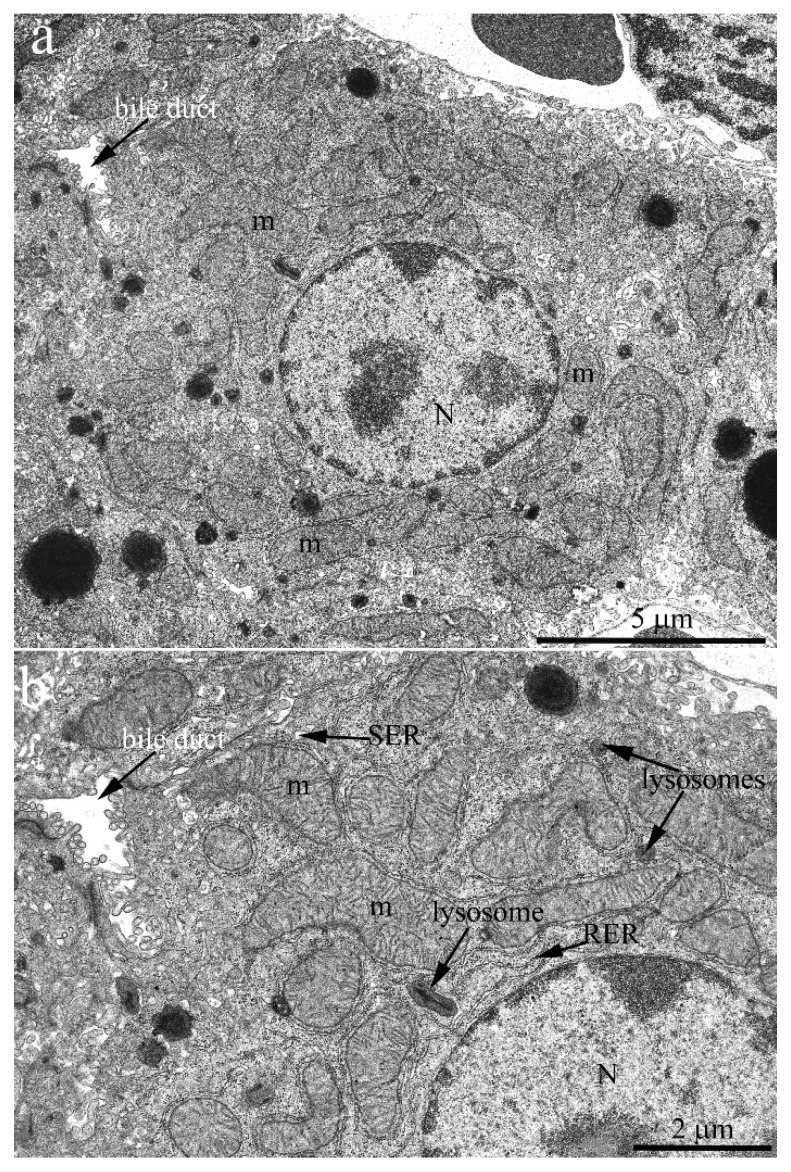
Hepatocyte ultrastructure in C57BL/6 mice aged 2 months. (**a**) Overview photograph. The hepatocyte has an irregular hexagonal shape with indistinct angles. A rounded nucleus (N) is located in the central part of the hepatocyte. The internal cell space is evenly filled with cell organelles. The arrow shows the bile duct; (**b**) the internal hepatocyte ultrastructure. In the hepatocyte cytoplasm, there are many mitochondria (m). Rough ER (RER) takes the form of stacks of membranes lying in parallel. Smooth ER (SER) membranes are minor and are located mainly close to the plasma membrane. Lysosomes are present in the cytoplasm.

**Figure 2 ijms-23-09067-f002:**
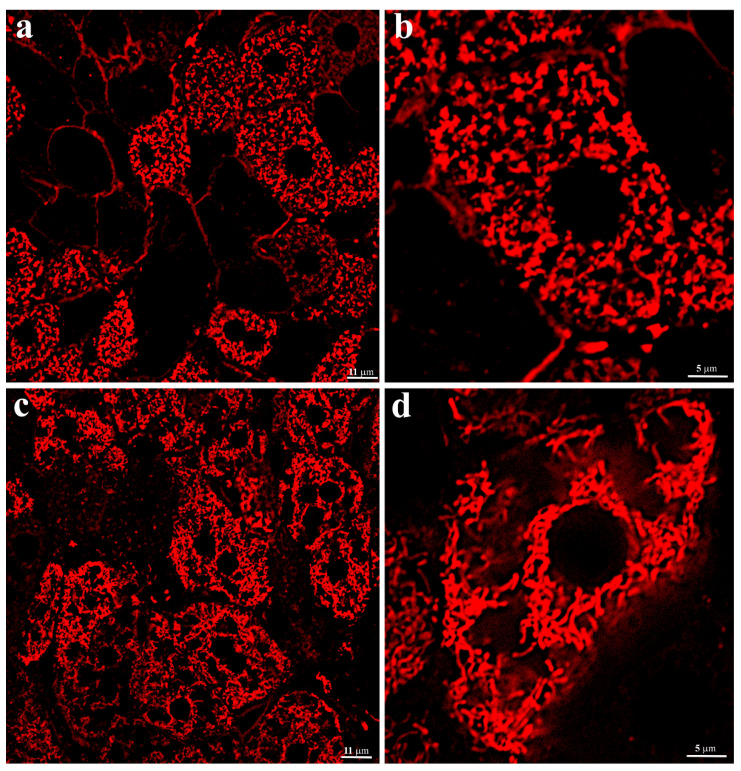
Evaluation of the transmembrane potential and structure of mitochondria in the vital liver tissue sections. Confocal microscopy of a vital liver tissue section stained with TMRE: (**a**) 2-months-old C57BL/6 mice, overview; (**b**) 2-months-old C57BL/6 mice, higher magnification; (**c**) 3-years-old *H. glaber*, overview; (**d**) 3-years-old *H. glaber*, higher magnification.

**Figure 3 ijms-23-09067-f003:**
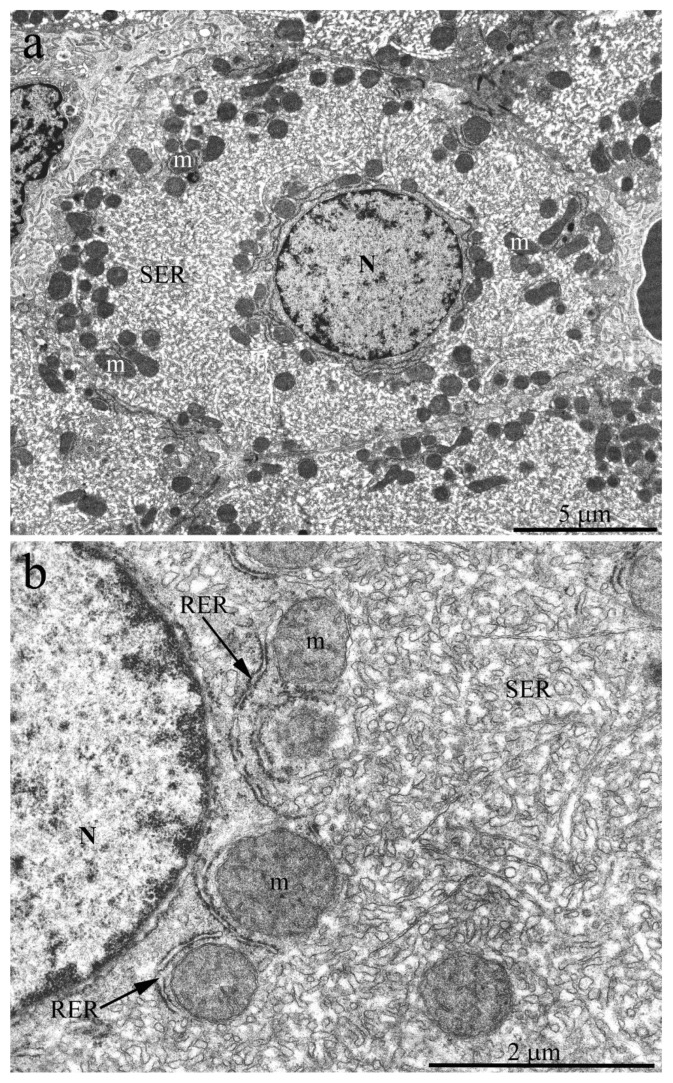
Hepatocyte ultrastructure of *H. glaber* aged 3 years. (**a**) Overview photograph. The nucleus (N) is located in the center of a polygonal cell. Numerous rounded or oval-shaped mitochondria (m) are located around the nucleus and on the periphery of cells. The rest of the internal hepatocyte space is densely filled with SER membranes; (**b**) Features of the internal hepatocyte ultrastructure. Most of the cytoplasm is densely filled with SER membranes, represented by a system of tubules and cisterns. RER is located mainly around the mitochondria (m). On its outer membrane there are numerous osmiophilic granules of 12–15 nm in diameter, i.e., ribosomes.

**Figure 4 ijms-23-09067-f004:**
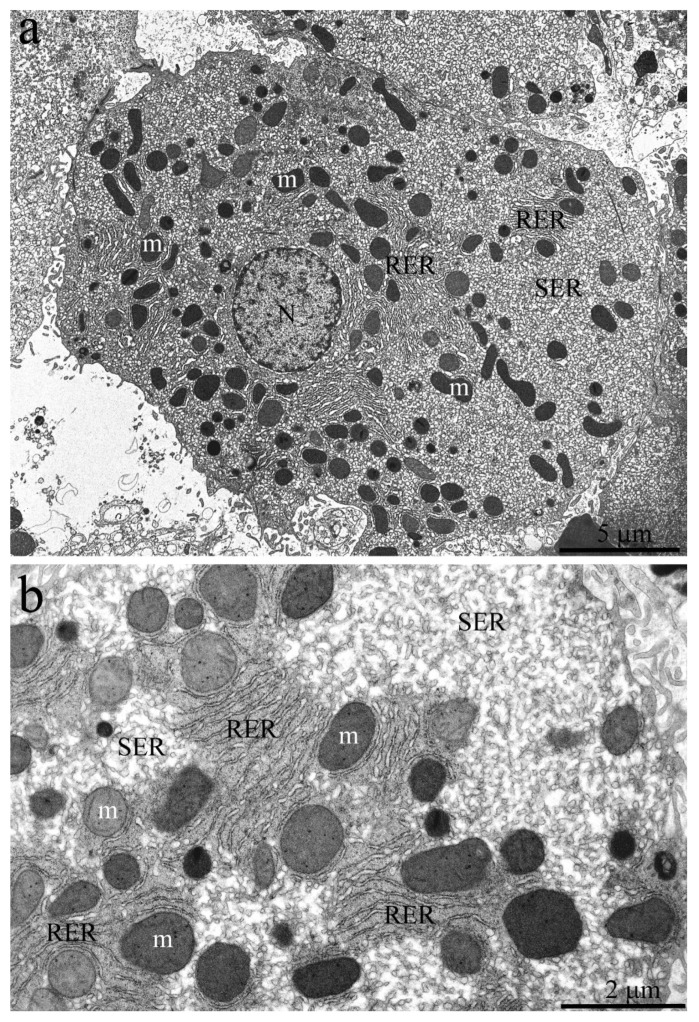
Hepatocyte ultrastructure of *H. glaber* aged 1 week. (**a**) Overview image of the hepatic cell ultrastructure. The nucleus (N) is located in the center of the cell. Numerous mitochondria (m), ER membranes (RER and SER), and other organelles are evenly distributed in the cytoplasm; (**b**) Features of the internal hepatocyte ultrastructure. The cells contain numerous stacks of RER, in close contact with mitochondria (m). Large areas of the cytoplasm are occupied by SER membranes.

**Figure 5 ijms-23-09067-f005:**
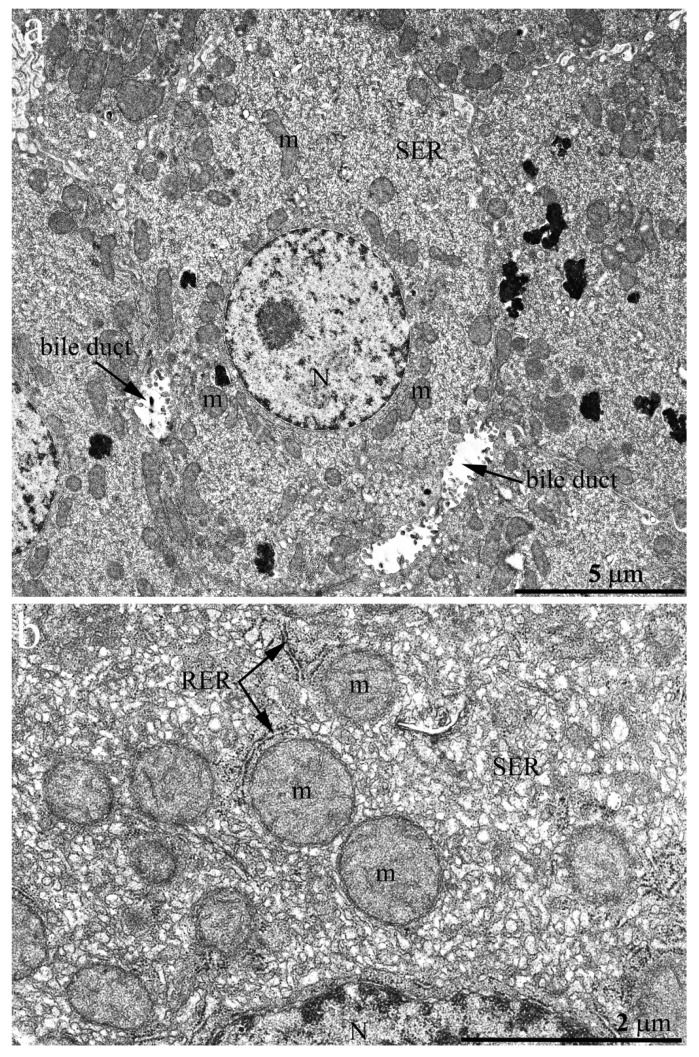
Hepatocyte ultrastructure of *H. glaber* aged 7 years. (**a**) Overview photograph. SER membranes tightly fill the cell volume. The area occupied by RER membranes is decreasing; (**b**) Features of the hepatocyte internal ultrastructure. There are no stacks of RER membranes—single cisterns are located between mitochondria (m). The entire internal space of the cells is filled with SER membranes.

**Figure 6 ijms-23-09067-f006:**
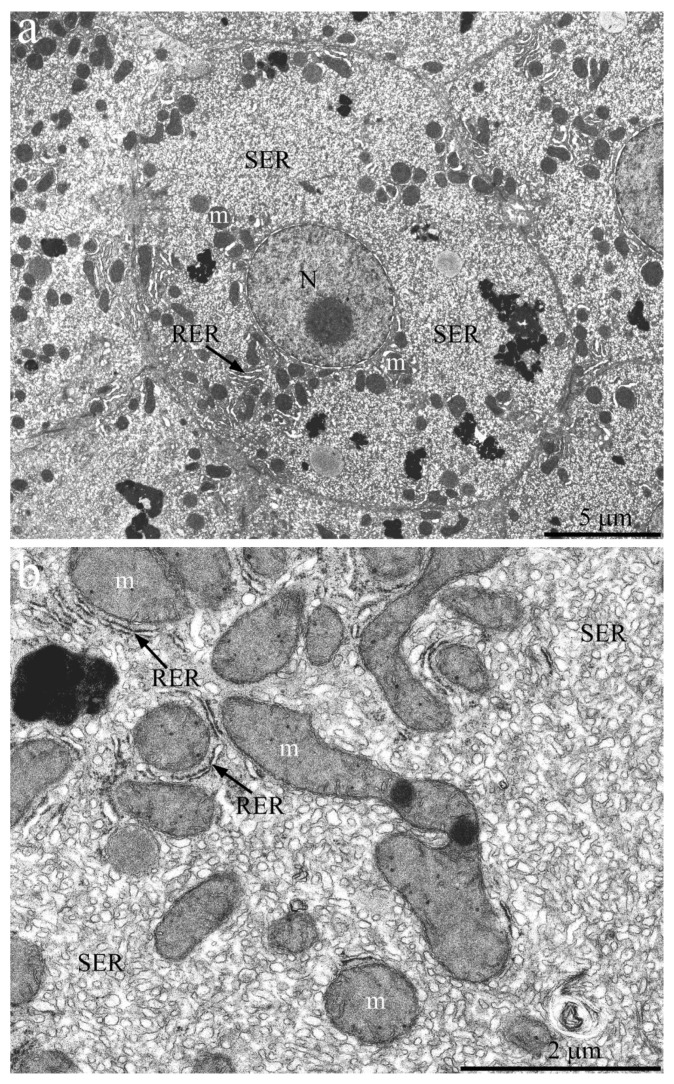
Hepatocyte ultrastructure of *H. glaber* aged 11 years. (**a**) Overview photograph. The nucleus (N) is located in the center of the cell; most of the cytoplasm is occupied by SER membranes. RER membranes are minor and surround mitochondria (m); (**b**) Features of the internal hepatocyte ultrastructure. Mitochondria (m) in cells are not evenly distributed but are localized in groups mainly in the perinuclear area and in the internal space of the cytoplasm. RER in cells is represented by single cisterns located between mitochondria.

**Figure 7 ijms-23-09067-f007:**
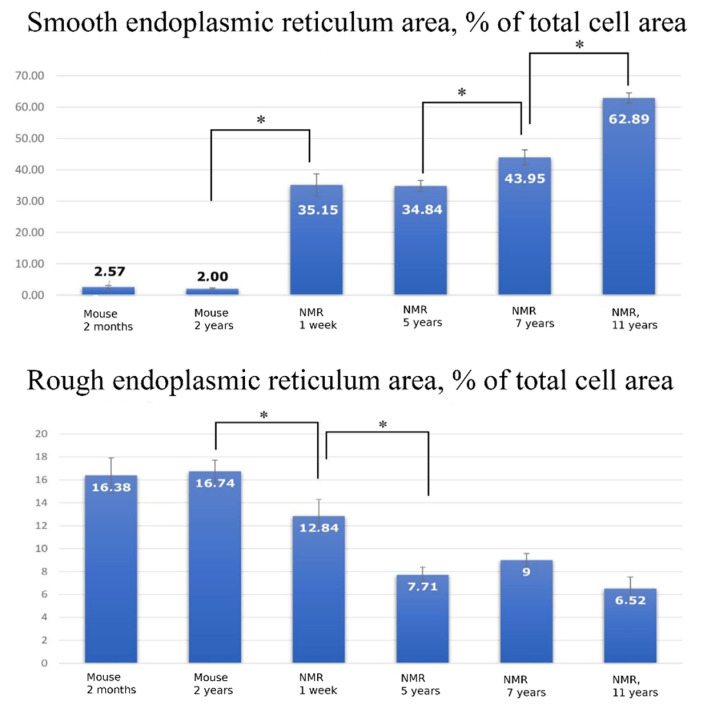
(**Top**) Proportion of SER area relative to total cell area (%) in the hepatocytes of C57BL/6 mice aged 2 months and 2 years, and *H. glaber* aged 1 week, 5 years, 7 years, and 11 years. (**Bottom**) Proportion of RER area relative to the total cell area (%) in the hepatocytes of C57BL/6 mice aged 3 months and 2 years, and *H. glaber* aged 1 week, 5 years, 7 years, and 11 years. * The difference is significant at *p* < 0.05. Error bars on all the graphs correspond to the standard error.

## Data Availability

The data presented in this study are available on request from the corresponding authors.

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
