# Peer review of "Unique Features of the Tissue Structure in the Naked Mole Rat (Heterocephalus glaber): Hypertrophy of the Endoplasmic Reticulum and Spatial Mitochondrial Rearrangements in Hepatocytes"

_ijms, 2022, doi:10.3390/ijms23169067_

Round 1

Reviewer 1 Report

The manuscript is well written and the experimental design is well organized. 

To date, so much effort is directed to the study of mitochondria, as cell power house. Here, the authors reported novel and appealing observations regarding the the spatial disposition and function of mitochondria in hepatocytes of a model of naked mole rat (Heterocephalus glaber). 

Such findings are deemed of interest because they raise important questions about mitochondrial respiration in the presence of low oxygenation and possible adaptation to hypoxia. I found this manuscript intriguing and full of curiosities, so much so it is deemed of interest in the journal.

The electron microscopic examination showed that H. glaber has a completely different pattern of organelle distribution in hepatic cells compared to other animals, open the possibility that hepatocytes adapt the morphology and the quantity of intracellular structures in response to external cues, throughout the life. 

Only minor points:

-did the authors observe alterations in mitochondrial structures?

-could the authors hypothesize possible translational implications of their observations?

-check the typos in the manuscript i.e. page 6 line 172 thorough or page 13 line 379 ImaJ.

-the quality of graphs should be ameliorated.

Author Response

We are very grateful to the reviewer for the positive assessment of our work. Below, point by point, we give our answers.

-did the authors observe alterations in mitochondrial structures?

No, the ultrastructure of mitochondria in both types of rodents did not show any visible changes. This fact has been added to the manuscript.

-could the authors hypothesize possible translational implications of their observations?

This is a very serious issue. Now we are still in the phase of accumulating facts about the differences in the structure and functions of cells and their elements in the naked mole rat and mouse, given that the first belongs to long-lived species, and the second to short-lived. In principle, in the end, firstly, if there are absolutely proven significant differences in life parameters, we will be able to predict to some extent the life expectancy of a particular species, and secondly, we will indicate the direction of changes that could ensure an increase in life expectancy. But so far this is a subject of the distant future and we are not able to speculate with a minimum of knowledge of the distinctive features of long-lived species, so we decided not to introduce such ideas into the manuscript.

-check the typos in the manuscript i.e. page 6 line 172 thorough or page 13 line 379 ImaJ.

Fixed. Thank you

-the quality of graphs should be ameliorated.

We understand the concern of the reviewer, but could not fix it so far since in case of including into our manuscript all graphs with higher resolution, the size of the file is very large exceeding the permitted size to submit the manuscript so we had to manipulate with the resolution to fit the permitted size of the file. We will talk to the journal if permitted for extracting the high resolution graphs from the data cloud which will be given. The access to the files is here:

https://disk.yandex.ru/d/TzH0rEIazSQj7A

Again, we would like to thank you for your work

Sincerely,

Dmitry B. Zorov,

Professor of Biochemistry

Reviewer 2 Report

This manuscript presents electron microscopic and confocal microscopy studies of the internal structure of hepatocytes of a unique organism (naked mole rat) in contrast to a standard warm-blooded animal (C57BL/6 mice) with a similar body weight to the first one. Of particular interest is the observed rearrangement of the RER/SER ratio with age in naked mole rats. It is possible that animals exposed to high CO2 concentrations have the effect of some pharmacological post-conditioning which may also contribute to the mild uncoupling effect. It would be nice to test ROS production levels in a similar way to the work done with TMRM (Figure 2). I would like to hope that these authors will continue this work in this direction. However, the shortcomings do not detract from the importance of this study. This manuscript can be published in IJMS after correction of existing comments which highlighted in yellow .

Author Response

We are very grateful to the reviewer for the positive assessment of our work. All changes which have been marked by yellow have been modified and thank you for pointing this. In the revised version our changes are marked with red.

Again, we would like to thank you for your work

Sincerely,

Dmitry B. Zorov,

Professor of Biochemistry
